# Changes in Molar Tipping and Surrounding Alveolar Bone with Different Designs of Skeletal Maxillary Expanders

**DOI:** 10.3390/biomedicines11092380

**Published:** 2023-08-25

**Authors:** Javier Echarri-Nicolás, María José González-Olmo, Pablo Echarri-Labiondo, Martín Romero

**Affiliations:** 1Doctoral Program in Health Sciences, International PhD School, Rey Juan Carlos University (URJC), 28922 Alcorcón, Spain; j.echarri.2020@alumnos.urjc.es; 2Department of Orthodontics, Rey Juan Carlos University, 28922 Alcorcón, Spain; martin.romero@urjc.es; 3Athenea Dental Institute, San Jorge University, 50830 Zaragoza, Spain; echarri@centroladent.com

**Keywords:** rapid maxillary expansion, maxillary skeletal expander, cone-beam computed tomography (CBCT), miniscrew-assisted rapid palatal expansion (MARPE), bone-anchored maxillary expander (BAME)

## Abstract

This study compared the buccolingual angulation (BLA) of the upper and lower first permanent molars before and after using the different methods of microimplant-assisted expansion in adults and its influence on bone insertion loss. Methods: Cone-beam computed tomography scans taken before and after the expansion in 36 patients (29.9 ± 9.4 years) were used to assess dental and periodontal changes and compare changes between the groups. Results: This research shows a statistically significant increase in the BLA of the upper first molars. An increase of the BLA of the lower molars is also observed in MARPE. Regarding the comparison between cases treated with MARPE (4.42° ± 10.25°; 3.67° ± 9.56°) and BAME (−0.51° ± 4.61°; 2.34° ± 4.51°), it was observed that upper molar torque increased significantly less in cases treated with BAME. In cases with CWRU < 96° at T0, a slight bone insertion gain was observed at T1, whereas if CWRU ≥ 96°, a slight bone insertion loss was observed. Regarding the labial cortical bone loss, a slight gain of CBW was observed in all cases. This labial cortical enlargement (T0–T1) is greater in cases where the CWRU < 96° at T0. Conclusions: Patients treated with MARPE show torque increase in the teeth selected to support the expansion appliance compared to cases treated with BAME. In cases where the BLA at T0 < 96°, an increase in thickness and cortical insertion is observed in the upper molars after treatment with disjunction appliances assisted with microscrews.

## 1. Introduction

Non-surgical rapid palatal expansion (RPE) in adults can provoke purely orthodontic expansion, i.e., undesired dentoalveolar effects [1] This limitation is due to the mid-palatal suture ossification, which makes the orthopedic separation of hemimaxillary portions impossible with the use of tooth-borne separators [2]. An expansion with increased torque and descent of the palatine cusps occurs in the upper posterior teeth [2]. In addition, adverse effects have been found, such as gingival inflammation, labial gingival recession, labial fenestration, and root resorption, due to their contact with the labial cortical bone [3].

The use of this technique is contraindicated in adults with maxillary compression, correct or positive molar torque, periodontal disease, generalized recessions, or absence of posterior teeth. If treatment is required, a surgical approach with surgical assisted rapid palatal expansion (SARPE) should be chosen. In SARPE, a surgical separation of the already ossified midpalatal suture is performed. In this way, we can increase the skeletal transverse dimension with an intraoral expansion device [4].

The microimplant-assisted rapid palatal expansion (MARPE) technique described by Moon [5] is characterized by a reduction of the excessive load exerted by conventional appliances to the labial periodontal ligament of the teeth, which is used as anchorage. The technique consists of a tooth-bone-borne device that uses retention in the form of two or four bicortical microscrews placed in the posterior area of the palate. The bone-anchorage maxillary expansion (BAME) appliance has been described as consisting of two or four bicortical microimplants without dental support [6]. 

The main side effect of MARPE is the buccolingual angulation (BLA) of the posterior teeth [7]. Increased molar torque has also been related to bone dehiscence [8]. Few studies have studied the change of BLA of the molars after treatment with MARPE or BAME using CBCTs [7,9,10,11,12].

Complications with microscrew-assisted separators occur in patients who have already completed the midpalatal suture ossification when the orthopedic forces are applied near the maxillary, frontal maxillary, and frontal zygomatic-maxillary buttresses. These buttresses distribute the applied force. There are reports of cases with fractures in the frontonasal suture or ocular area. In addition, there have also been cases with device impaction or microscrews fracture that can be solved only by a surgical intervention [13,14,15].

The BLA of the upper and lower molars forms the curve of Wilson (COW), which has a convex shape in the upper posterior teeth and a concave shape in the lower posterior teeth [16]. According to Hayes [16], the correct BLA of both the canines and the upper and lower molars is assumed to be essential to ensuring the functional balance of the patient and the stability of orthodontic treatment. The objective of having correct molar occlusion are twofold: (i.) the centering and uprighting of the teeth in their alveolus and (ii.) ensuring their correct intercuspation. These factors are important for achieving the proper functioning of dentition, dental stability, temporomandibular joint health, and periodontal viability [17].

The correct relationship of COW has been studied in treatments performed with fixed appliances [18,19,20,21] and with aligners [22], but it has not been studied in treatments performed with MARPE or BAME. 

The literature also describes how cortical bone insertion (CBI) affects the molars in MARPE or BAME treatments [7,23,24,25], but no relation has been found with the increased torque of the affected teeth. 

This article has two main objectives. The first is to analyze and compare the change of the BLA of the upper and the lower first permanent molars before and after using the different methods of microimplant-assisted expansion (MARPE and BAME). The second is to evaluate whether the initial buccolingual inclination of the upper first molars affects periodontal health after MARPE treatment.

This article has great clinical significance for orthodontists since it provides rigorous information on which expansion therapy has the most dental effect on the maxillary and mandibular molars, and it is important to evaluate the inclination of the upper molars before treatment for the control of periodontal health.

## 2. Materials and Methods

### 2.1. Design and Participants

Patients’ data were involved retrospectively. Those patients under treatment with MARPE or BAME in a private dental office (Athenea Dental Institute, Barcelona, Spain) from September 2021 to March 2023 were involved. 

The inclusion criteria of the study were adult patients with maxillary compression without counter-indications for surgery and who were to undergo microimplant-assisted maxillary expansion treatment. The exclusion criteria from the study were patients with craniofacial malformations, patients with fissured palates, patients who did not accept orthodontic treatment, and patients who refused to participate in the study or refused to sign the informed consent. Demographics and sample images were used, and the information was anonymized. The sample size was calculated using Jamovi 2.3.18. As there were no previous studies comparing dental changes using BAME and MARPE, sample size calculations were based on the results of a pilot study performed with ten patients. The calculated means ± standard deviations (SDs) of the maxillary width change of BAME and MARPE were 1.96 mm ± 0.22 and 2.31 mm ± 0.35, respectively. Based on comparison of means using the two-tailed test, it was calculated that accepting an alpha risk of 0.05 and a beta risk of 0.2 in a bilateral contrast, 16 subjects per group are required to detect a difference equal to or greater than 0.35 mm. Finally, we were able to include 36 subjects, which increased the robustness of the data. The MARPE technique was used in 18 subjects, and the BAME technique was used in 18 subjects. See Figure 1 and Figure 2.

### 2.2. Digital Procedure and Measurements in the CBCT

Palalign Round Head Type (Osteonic Co., Ltd., Seoul, Republic of Korea), an Ti6Al4V alloy, with diameter of 1.8 mm diameter and 10, 12, 14, or 16 mm long (depending on the case) microimplants are used to ensure bicorticality. It has been described that this bicorticality is mandatory to ensure the increase the stability and decrease mini-implant deformation and fracture [25]. All appliances were designed digitally, and all the microscrews were placed with digital guidance to reduce clinical placement error [26]. A Power MARPE Type 1 (Osteonic Co., Ltd., Seoul, Republic of Korea) expansion screw was used with an activation rate of four turns per day until the interincisal diastema appeared and then, two turns per day until a 1.5 mm overcorrection per side is achieved. All treatments were performed by the same orthodontist. The study was conducted according to the guidelines of the Declaration of Helsinki. The protocol was reviewed and approved by the Ethics Committee of the Rey Juan Carlos University with internal number (1504202110721).

#### 2.2.1. CBCT Data Acquisition

The patient was subjected to a CT-type radiographic recording (NewTom Giano HR (QR, Verona, Italy) with 300 μm voxel size and a 16 × 18 cm fov before and after MARPE or BAME treatment, and the following indicators were calculated on that 3D X-ray before (T0) and after treatment (T1). All treatments were performed by the same orthodontist.

#### 2.2.2. CBCT Measurement

For the CBCT BLA measurements, the transverse analysis of Case Western Reserve University (CWRU) was followed [27]. Maxillary first molar BLA was measured in the coronal view as the angle between the major axis of the palatal root and the mesiopalatal cusp and the line tangent to the nasal floor. The BLA of the lower molars was the angle between the mesial cusp and the mesial root and the tangent line at the lower edge of the mandibular corpus [28]. The indicators are shown in Table 1 and Figure 3.

Cortical bone insertion (CBI) and cortical bone width (CBW) of the right and left upper first molars were also measured (Figure 3). 

An example of how the measurements were carried out can be found in the Appendix A.

### 2.3. Statistical Analysis

All statistical analyses were performed using the Statistical Package for the Social Sciences version 28.0 for Windows (IBM, Armonk, NY, USA). The Kolmogorov–Smirnov test was used to evaluate the assumption of normality, which was confirmed. A descriptive analysis was carried out to expose the details of the sample, such as age, sex, appliance type, maturation stage of the suture, and type of suture opening achieved after treatment. An intraclass correlation coefficient (ICC) was used to determine the intra-observer reliability of the measurements through reliability analysis in SPSS. The intraclass correlation coefficient (ICC) was calculated considering poor (ICC < 0.40), fair to good (0.40 ≤ ICC ≤ 0.75), and excellent (ICC > 0.75). Three measurements were also carried out for each indicator and for each investigation time, and the measurement error was calculated. Subsequently, a paired sample *t*-test was performed to evaluate the T0–T1 change of dental measurements obtained with MARPE and BAME, and a student *t*-test to compare changes in skeletal measurements between MARPE and BAME. In addition, Cohen’s d was used for the effect of the sample in the analysis of the differences of the means with the *t*-test. A measurement with low effect was considered as d ≈ 0.2, medium d ≈ 0.5, and high d ≈ 0.8 [29]. An ANOVA one-factor analysis was performed to evaluate the loss of CBI and CBW after treatment with both therapies. Then, a univariate analysis was done to evaluate the loss of CBI and CBW and their relationship with the CWRU at T0. Statistical significance was established at *p* < 0.05.

## 3. Results

This study presents an ICC > 0.9 in the evaluations performed (Table 2).

### 3.1. General Descriptive Analysis

The sample size was 36 subjects, and 41.7% of the subjects were male and 58.3% were female. The mean age of the subjects was 27.42 ± 8.53, with an age range of 18 to 49. The MARPE technique was used in 18 subjects (50%), and the BAME technique was used in 18 subjects (50%). Regarding the relationship between device type and sex, MARPE was used in 60% of the male subjects, whilst BAME was used in the remaining 40%, and MARPE was used in 42.9% of the female subjects, whilst BAME in the remaining 57.1%. No differences in gender distribution were found (X(1) = 1.029; *p* = 0.310).

### 3.2. Evaluation of Variables Baseline

To evaluate the similarity between groups at T0, age was compared (MARPE: 29.9 ± 9.4 and BAME: 24.8 ± 6.8; t = 1.83, *p* = 0.076) finding no significant differences. The clinical variables (See Table 3) are right cortical bone insertion (MARPE: 2.9 ± 1.3 y BAME: 2.9 ± 1.7; t = 0.01, *p* = 0.496) and left (MARPE: 3.1 ± 0.9 y BAME: 2.8 ± 1.2; t = 0.667, *p* = 0.255), right cortical bone width (MARPE: 6.66 ± 1.6 y BAME: 7.1 ± 0.8; t = 1.044, *p* = 0.153) and left (MARPE: 6.8 ± 1.0 y BAME: 7.1 ± 0.9; t = 0.947, *p* = 0.175), buccal lingual angulation in right upper molar (MARPE: 99.5 ± 7.3 y BAME: 101.4 ± 6.4; t = 0.829, *p* = 0.413), and buccal lingual angulation in left upper molar (MARPE: 97.4 ± 7.8 y BAME: 102.8 ± 10.1; t = 0.1.782, *p* = 0.08). No differences were found between either groups.

### 3.3. Changes in T1–T0 Bucolingual Angulation of First Molar

The angulations of the upper and lower right molars as well as the left first molars were calculated according to the expansion therapy used. Torque increases of 4.42 ± 10.25° and 3.67 ± 9.56° in the right and left upper first molars, respectively, were observed in the cases treated with MARPE. Torque increases of − 0.51 ± 4.61° and 2.34 ± 4.51° in the right and left upper first molars, respectively, were observed in the cases treated with BAME. Compared with the used therapies (Table 4), MARPE produced a greater bucolingual inclination compared to BAME, although these differences were not statistically significant.

In relation to BLA 36 and BLA 46, there was an increase in BLA for the cases treated with MARPE (BLA 36: 1.02 ± 4.21°; BLA 46: 1.12 ± 6.04°), but for BAME, this increase was lower than MARPE for only one of the lower molars (BLA 46: 0.79 ± 4.17°), whereas for BLA 36, there was an inclination loss (−1.1 ± 6.71°) (see Table 4).

### 3.4. The Loss of CBI and CBW and Their Relationship with the CWRU at T0

In cases where the CWRU was less than 96° at the beginning of the treatment, a slight bone insertion gain (0.24 ± 0.61 mm) was observed, whereas if the CWRU was 96–104° (−0.08 ± 1.05 mm) or higher than 104° (−0.29 ± 1.06 mm), a slight bone insertion loss was observed. These differences were not significant (F = 0.874, *p* = 0.427), as seen in Figure 4.

Regarding labial cortical bone loss, a slight loss was observed in all the groups, but the least loss was observed in the CWRU = 96–104° group (−0.01 ± 0.69 mm). There was a greater loss for the CWRU > 104° group (−0.24 ± 0.61 mm) and a median loss for the CWRU < 96° group (−0.13 ± 0.53 mm).

This loss of labial cortical bone was less in cases where the posterior molar torque was correct at T0, but these differences were not significant (F = 0.440, *p* = 0.648), as seen in Figure 5.

## 4. Discussion

The two main objectives of this research were to study (i.) the BLA changes in the first upper and lower molars and (ii.) the loss of CBI and CBW in patients with maxillary compression treated with MARPE and BAME. For this study, different measurements were taken in the CBCT, and they were taken before (T0) and after (T1) maxillary expansion therapy. Several studies, such as those by Shewinvanakitkul [27], Evangelinakis [30], Shewinvanakitkul et al. [31], Karamitsou [32], Miyamoto [33], Copeland [34], Streit [35], and Palomo et al. [36], have described the CWRU method for measuring the BLA in canines and the upper and lower molars, and these studies have established some standards. Those standards were (i) 100 ± 4° of BLA in upper first molars and (ii) 77 ± 4° of BLA in lower first molars. 

The labial molar angulation after MARPE therapy is commonly found in the literature [8,37,38,39], but it has not been compared with the labial angulation of the lower molars or in treatments performed with MARPE and BAME techniques.

In the article by Jia et al. [38], dental changes were compared in 30 patients treated with MARPE and 30 patients treated with conventional bone-borne maxillary expansion. CBCTs were taken before expansion and 1 week after expansion, and the change of the labiolingual inclination of the upper right and left first molars was studied, which resulted in the obtainment of the angle between the lines passing through the palatal orifice of the chamber and the apex of the palatal root and the original horizontal line. Torque increases of 3.82 ± 4.07° and 2.72 ± 3.44° were shown in the right and the left upper first molars, respectively. The results obtained in our research for the cases treated with MARPE do not coincide with the results obtained in the article by Jia et al. [38], and this disparity may be because our research was conducted in adults. 

De Oliveira et al. [39] compared skeletal and dental changes in 17 patients treated with MARPE and 15 patients treated with SARPE. Three-dimensional radiographic records were also taken just before and after the expansion. In the group of patients treated with MARPE, the increase in BLA of the upper right and left first molars was 2.87 ± 1.94° and in the SARPE group, it was 3.39 ± 2.41°.

Gunyuz Toklu et al. [37] compared periodontal, dentoalveolar, and skeletal changes in cases treated with tooth-borne and tooth-bone-borne expansion appliances in 26 patients. CBCTs were taken before treatment (T1) and at the end of 3 months of retention (T2). It measured the changes in the molar mesiobuccal bone thickness (M1 MBBT) and molar dental inclination (M1 D1) of the upper right and left first molar between T1 and T2. The results of the MARPE group for M1 MBBT were 0.89 ± 0.72 mm and 0.98 ± 0.61 mm and for M1D1 were −2.43 ± 5.54° and −2.89 ± 4.90° for right and left upper first molar, respectively, between T1 and T2. The results obtained in our research for the cases treated with MARPE do not coincide with the results in the article of Gunyuz Toklu et al. [37], and this disparity may be because our second measurement in the CBCT was immediately after expansion.

This research shows an increase in the BLA of the upper first molars. A variation in the BLA of the lower molars was also observed. Regarding the comparison between the cases treated with MARPE and BAME, it was observed that the molar torque increased significantly less in cases treated with BAME. This increase may be due to the dental anchorage at the level of upper first molars of the design in MARPE appliances. These results contradict the conclusions drawn by Khosravi et al. [40] in their systematic review of tooth tipping in treatments carried out using MARPE and BAME devices. The article by Khosravi et al. [40] compared four studies that examined rapid maxillary expansion between bone-borne and tooth-borne appliance following surgical assisted maxillary expansion. Similar dental tippings were reported for both bone-borne and tooth-borne appliances with no significant difference between the two devices.

Whether the initial BLA in the upper first molars at T0 affected the loss of CBI and CBW was also considered. It was observed that when the BLA (100 ± 4° according to CWRU) at T0 was correct, there is increased bone insertion. In cases where the buccolingual angulation did not fall within the CWRU standard, slight bone insertion loss was observed. The gains were slight, but there were differences. Regarding the CBW loss, CBW gain was observed. If the T0 BLA was negative and MARPE or BAME therapies were performed, the labial cortical bone gained some width. These differences were not statistically significant. The results of this research coincide with those previously observed in the study by Baysal et al. [41] regarding CBI loss, but our findings do not support the conclusions drawn in a previous study [41] that also showed CBW loss.

The contribution of this study should be evaluated after taking into account its limitations. The data presented in this research should be interpreted with caution due to the limited sample size. The observation period of the results was also short, and a measurement after the removal of the device may prove useful to corroborate the stability of the results. As for the periodontal measurements, an investigation in which they will be performed using a periodontal probe instead of measurements taken from CBCTs would also be beneficial.

Other variables, such as root resorption of the first molars, were not taken into account. These variables have been assessed in the literature [42], demonstrating root resorption even in teeth not anchored to the appliance; however, this resorption was lower in cases treated with skeletal-supported appliances [43].

The results of this study have a number of important implications for the future practice of orthodontists. On the one hand, the microimplant-assisted expander will bring about a series of dentoalveolar changes in the upper molars. The choice between MARPE and BAME will play an important role in the amount of torque increase as well as the CBI and CBW loss in the upper molars.

## 5. Conclusions

Patients treated with MARPE show torque increases in the teeth selected to support the expansion appliance compared to cases treated with BAME.In cases where the upper molar torque was correct at the beginning of the treatment, a slight bone insertion gain was observed.In cases where the upper molar torque was negative or positive, a slight bone insertion loss was observed.Regarding the labial cortical bone loss, a slight gain of CBW was observed in all cases. This increase in labial cortical is higher in cases where posterior molar torque was negative before treatment.

## Figures and Tables

**Figure 1 biomedicines-11-02380-f001:**
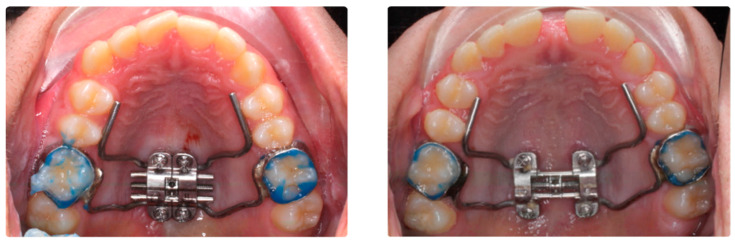
Pre- and post-expansion MARPE device design.

**Figure 2 biomedicines-11-02380-f002:**
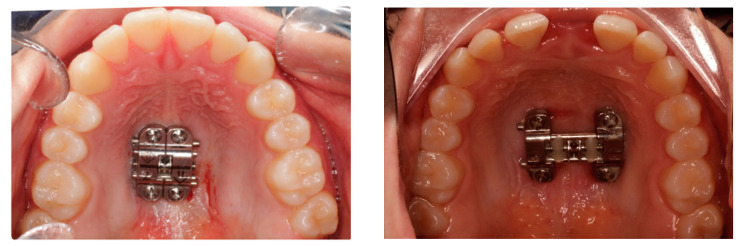
Pre- and post-expansion BAME device design.

**Figure 3 biomedicines-11-02380-f003:**
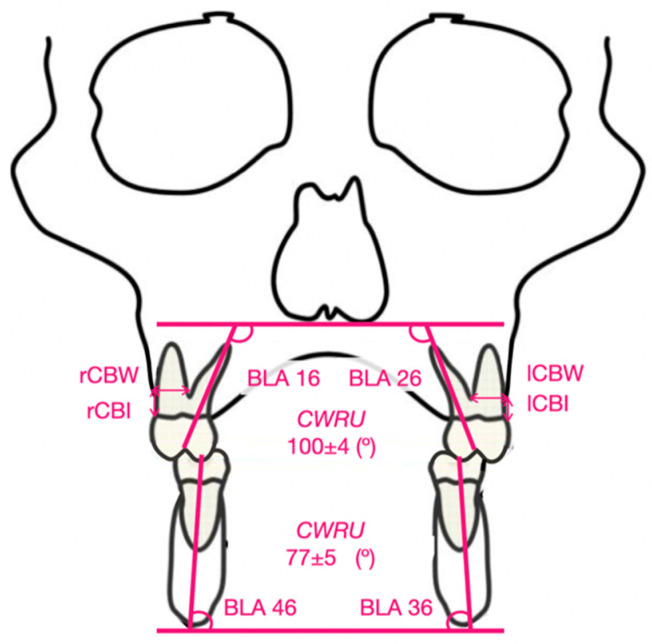
Indicators of BLA, CBI, and CBW. Angulation standard in the maxillary and mandibular molars according to CWRU.

**Figure 4 biomedicines-11-02380-f004:**
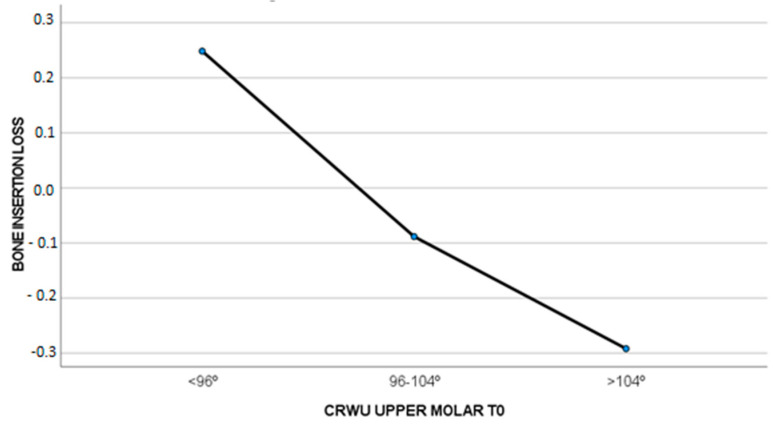
Amount of CBI loss (mm) compared to CWRU T0.

**Figure 5 biomedicines-11-02380-f005:**
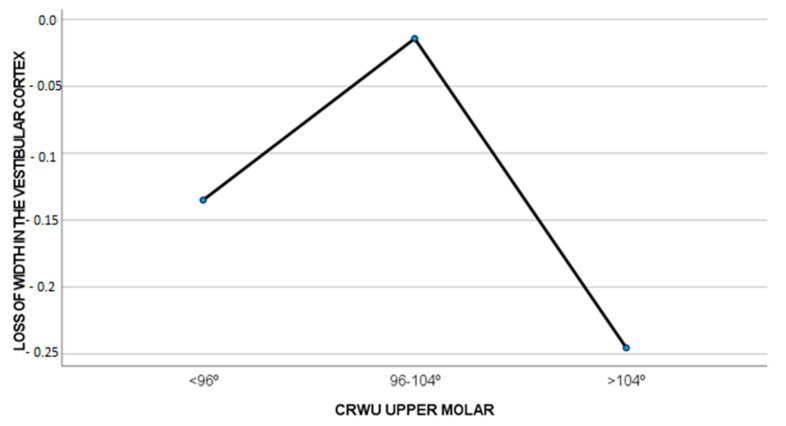
Amount of CBW loss (mm) compared to CWRU T0.

**Table 1 biomedicines-11-02380-t001:** Description of indicators (BLA: buccolingual angulation, CBI: cortical bone insertion, and CBW: cortical bone width).

Measurements	Description
Bucolingual angulation of the first right upper molar (BLA 16)	Angle between the line passing through the palatal cusp and the root apex of the palatal root of the right upper first molar and the tangent line to the floor of the nostrils.
Bucolingual angulation of the first left upper molar (BLA 26)	Angle between the line passing through the palatal cusp and the root apex of the palatal root of the left upper first molar and the line tangent to the floor of the nostrils.
Bucolingual angulation of the first left lower molar (BLA 36)	Angle between the line passing through the central fossa and the root apex of the lingual root of the first right lower molar and the line tangent to the lower edge of the mandibular corpus.
Bucolingual angulation of the first right lower molar (BLA 46)	Angle between the line passing through the central fossa and the root apex of the lingual root of the first lower left molar and the line tangent to the lower edge of the mandibular body.
Cortical bone insertion (CBI)	Labial cortical bone insertion with respect to the cementoenamel junction of the upper first molars
Cortical bone width (CBW)	Labial cortical bone thickness. The distance from the furca of the upper first molars to the most external point of the labial cortical bone was taken.

**Table 2 biomedicines-11-02380-t002:** Intraclass Correlation Coefficient (ICC) and Measurement Error (ME).

	ICC	ME
BLA 16	0.972	0.26° ± 0.56
BLA 26	0.912	0.19° ± 0.42
BLA 36	0.998	0.37° ± 0.67
BLA 46	0.986	0.25° ± 0.55

**Table 3 biomedicines-11-02380-t003:** Descriptive and comparative analysis of changes in BLA for MARPE T1–T0 and descriptive and comparative analysis of changes in BLA for BAME T1–T0.

	MARPE T0 M (SD)	MARPE T1 M (SD)	MARPE T1–T0 M (SD)	*p* Value T1–T0
BLA 16 (°)	99.52 (7.35)	103.94 (8.02)	4.42 (10.25)	0.085
BLA 26 (°)	97.47 (7.89)	101.14 (7.69)	3.67 (9.56)	0.122
BLA 36 (°)	76.01 (5.50)	77.26 (5.95)	1.02 (4.21)	0.222
BLA 46 (°)	76.46 (8.13)	77.58 (6.93)	1.12 (6.04)	0.440
	BAME T0	BAME T1	BAME T1–T0	*p* value T1–T0
BLA 16 (°)	101.44 (5.88)	100.93 (8.82)	− 0.51 (4.61)	0.644
BLA 26 (°)	102.88 (10.16)	105.22 (8.75)	2.34 (4.51)	0.042 *
BLA 36 (°)	77.06(5.81)	75.96 (6.41))	− 1.10 (6.71)	0.496
BLA 46 (°)	77.57(7.61)	78.36 (6.51)	0.79 (4.17)	0.434

* = *p*< 0.05.

**Table 4 biomedicines-11-02380-t004:** Comparative analysis of T1–T0 for MARPE and BAME appliances.

	MARPE T1–T0	BAME T1–T0	*p* Value	Cohen’s d
BLA 16 (°)	4.42 (10.25)	− 0.51 (4.61)	0.071	0.620
BLA 26 (°)	3.67 (9.56)	2.34 (4.51)	0.594	0.177
BLA 36 (°)	1.02 (4.21)	− 1.10 (6.71)	0.274	0.378
BLA 46 (°)	1.12 (6.04)	0.79 (4.17)	0.847	0.063

## Data Availability

The data that support the findings of this study are available on request from the corresponding author. The data are not publicly available due to privacy and ethical restrictions.

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
