# Peer review of "Changes in Molar Tipping and Surrounding Alveolar Bone with Different Designs of Skeletal Maxillary Expanders"

_biomedicines, 2023, doi:10.3390/biomedicines11092380_

Round 1

Reviewer 1 Report

Authors need to have a English as first language person help edit the grammar of the manuscript. There is a sentence in Spanish. Also, authors mention 3 objectives but only list 2. Manuscript has potential but needs some editing. Add SD on the measurement errors. What is the age range of the patients?

Did the authors consider the clinical significance, what was it?

Other than that, manuscript does add info to dental effects of bone-anchored expanders which is very applicable to clinical practice now-a-days.

needs editing and one sentence needs to be translated

Author Response

Authors need to have a English as first language person help edit the grammar of the manuscript. There is a sentence in Spanish. 

We have sent the article to the translation service for editing and proofreading. 

Also, authors mention 3 objectives but only list 2. Manuscript has potential but needs some editing. 

This was corrected in the line 63.

Add SD on the measurement errors

This was corrected in the Table 2

What is the age range of the patients?

The mean age of the subjects was 27.42±8.53, with an age range of 18 to 49

Did the authors consider the clinical significance, what was it?

This article has a great clinical implication for orthodontists since it provides rigorous information on which expansion therapy has the most dental effect on the maxillary and mandibular molars. It is important to evaluate the inclination of the upper molars before treatment for the control of periodontal health.

The results of this study have a number of important implications for the future practice of orthodontists. On the one hand, the microimplant-assisted expander will bring about a series of dentoalveolar changes in the upper molars. The choice between MARPE and BAME will play an important role in the amount of torque increase and CBI and CBW loss in the upper molars.

Other than that, manuscript does add info to dental effects of bone-anchored expanders which is very applicable to clinical practice now-a-days.

Reviewer 2 Report

Dear Authors,

I’ve extensively read the manuscript titled “Changes in molar tipping and surrounding alveolar bone with 2 different design of skeletal maxillary expanders.”. The aim of this study is to compare the buccolingual angulation (BLA) of the upper and lower first per-12 manent molars before and after using the different methods of microimplant-assisted expansion in 13 adults and its influence on bone insertion loss. The methodology is appropriate and quite linear with recent evidences/ studies on this topic. I’ve not major concerns in this regard.

Some aspects must be adsdressed before considering the manuscript suitable for publication:

-       A significant revision of scientific English language is required.

-       CBCT images of measurements should be provided (at least as supplementary materials)

-       Authors should mention other side-effects related to molar tipping after maxillary expansion such as root resorption. Recent evidences using 3d technology have compared tooth-borne and bone-borne systems and found also interesting data in non-abutment teeth and should be reported or cited (https://pubmed.ncbi.nlm.nih.gov/36464753) (https://pubmed.ncbi.nlm.nih.gov/37573295/)

-       Authors should mention the importance of guided system when using other types of bone-borne expanders, citing appropriate references (https://pubmed.ncbi.nlm.nih.gov/37460998/)

-       In general, the discussion section should be improved, even considering the above mentioned topics

-       Was power analysis performed?

A significant revision of scientific English language is required.

Author Response

Dear Authors,

I’ve extensively read the manuscript titled “Changes in molar tipping and surrounding alveolar bone with 2 different design of skeletal maxillary expanders.”. The aim of this study is to compare the buccolingual angulation (BLA) of the upper and lower first per-12 manent molars before and after using the different methods of microimplant-assisted expansion in 13 adults and its influence on bone insertion loss. The methodology is appropriate and quite linear with recent evidences/ studies on this topic. I’ve not major concerns in this regard.

Some aspects must be adsdressed before considering the manuscript suitable for publication:

-       A significant revision of scientific English language is required.

We have sent the article to the translation service for editing and proofreading. 

-       CBCT images of measurements should be provided (at least as supplementary materials)

It was attached an example of the CBCT measurements of one patient as example.

-       Authors should mention other side-effects related to molar tipping after maxillary expansion such as root resorption. Recent evidences using 3d technology have compared tooth-borne and bone-borne systems and found also interesting data in non-abutment teeth and should be reported or cited (https://pubmed.ncbi.nlm.nih.gov/36464753) (https://pubmed.ncbi.nlm.nih.gov/37573295/)

It was added at the line 277 "Other variables, such as root resorption of the first molars, were not taken into account. These variables have been assessed in the literature [35] demonstrating root resorption even in teeth not anchored to the appliance; however, this resorption was lower in cases treated with skeletal-supported appliances [36].”

  1. Leonardi R, Ronsivalle V,Barbato E, Lagravère M, Flores-Mir C, Lo Giudice A. External root resorption (ERR) and rapid maxillary expansion (RME) at post-retention stage: a comparison between tooth-borne and bone-borne RME. Progress in Orthodontics, 2022;23(1)
                36. Leonardi R, Ronsisvalle V, Isola G, Cicciù M, Lagravère M, Flores-Mir C, Lo Giudice A. External root resorption and rapid maxillary expansion in the short-term: a CBCT comparative study between tooth-borne and bone-borne appliances, using 3D imaging digital technology. BMC Oral Health, 2023;558;23(1)

-       Authors should mention the importance of guided system when using other types of bone-borne expanders, citing appropriate references (https://pubmed.ncbi.nlm.nih.gov/37460998/)

It was added at the line 106 “All appliances were designed digitally and all the microscrews were placed digital-guided to reduce de clinical placement error [19]."

  1. Ronsivalle V, Venezia P, Bennici O, D’Antò V, Giudice A. Accuracy of digital workfloe for placing orthodontic miniscrews using generic and licensed open systems. A 3d imaging analysis of non-native .stl files for guided protocols. BMC oral health, 2023; 494, 23(1).

-       In general, the discussion section should be improved, even considering the above mentioned topics

This has already been answered in the previous points.

-       Was power analysis performed?

How the sample size calculation was performed is described in line 86-93.
"The sample size was calculated using Jamovi 2.3.18. As there were no previous studies comparing dental changes using BAME and MARPE, sample size calculations were based on the results of a pilot study performed in ten patients. The calculated mean ± standard deviation (SD) of the maxillary width change of BAME and MARPE was 1.96 mm ± 0.22 and 2.31 mm ± 0.35, respectively. Based on comparison of means, using two-tailed test, it was calculated that accepting an alpha risk of 0.05 and a beta risk of 0.2 in a bilateral contrast, 16 subjects per group are required to detect a difference equal to or greater than 0.35 mm. Finally, a sample was taken from 36 subjects"

Reviewer 3 Report

Dear Authors, 

The reviewer really appreciates the efforts of the authors to conduct this study which has a good clinical significance. However, there are some scopes to improve the quality of the manuscript. The reviewer would like to suggest the following revision in the manuscript to make it suitable for publication.

The abstract needs to be re-structured by shortening the methodology and adding highlighted results and a clear conclusion/ outcome of the study.

Some  old references need to be replaced with the recent study stating the fact.

The methodology needs to be revised in a more organized way by adding a sub-heading for each step for clear understanding.

The picture/ illustration of the methodology should be expressed in a single composite image/ illustration following the sequence.

Although the authors clearly mention the inclusion and exclusion criteria, the method of calculating the sample size is missing in the methodology

The discussion section is in lacks citations and a logical explanation of the result obtained in this study.

The conclusion section needs to be revised with a more clear and summarized outcome of the study.

Author Response

Dear Authors, 

The reviewer really appreciates the efforts of the authors to conduct this study which has a good clinical significance. However, there are some scopes to improve the quality of the manuscript. The reviewer would like to suggest the following revision in the manuscript to make it suitable for publication.

The abstract needs to be re-structured by shortening the methodology and adding highlighted results and a clear conclusion/ outcome of the study.

The new abstract section is “Abstract: This study compared the buccolingual angulation (BLA) of the upper and lower first permanent molars before and after using the different methods of microimplant-assisted expansion in adults and its influence on bone insertion loss.

Methods: Cone-Beam computed tomography scans taken before and after the expansion in 16 patients (29.9 ± 9.4 years) were used to assess dental and periodontal changes and compare changes between the groups.
Results: This research shows a statistically significant increase in the BLA of the upper first molars. A in-crease of the BLA of the lower molars is also observed in MARPE. Regarding the comparison between cases treated with MARPE (4.42º±10.25º; 3.67º±9.56º) and BAME (-0.51º±4.61º; 2.34º±4.51º), it was observed that upper molar torque increased significantly less in cases treated with BAME. In cases with CWRU<96º at T0, a slight bone insertion gain was observed at T1, whereas if CWRU≥96º, a slight bone insertion loss was observed. Regarding the labial cortical bone loss, a slight gain of CBW was observed in all cases. This labial cortical enlargement (T0-T1) is greater in cases where the CWRU<96° at T0.

Conclusions: Patients treated with MARPE show torque increase in the teeth selected to support the expansion appliance, compared to cases treated with BAME. In cases where the BLA at T0 <96º, an increase in thickness and cortical insertion is observed in the upper molars after treatment with disjunction appliances assisted with microscrews."

Some  old references need to be replaced with the recent study stating the fact.

It has been changed the reference:

Thilander B, Nyman S, Karring T, Magnusson I. Bone regeneration in alveolar bone dehiscences related to orthodontic tooth movements. Eur J Orthod. 1983;5(2):105–14.

to 

Kim H, Park SH, Park JH, Lee KJ. Nonsurgical maxillary expansion in a 60-year-old patient with gingival recession and crowding. Korean J Orthod. 2021;25;51(3):217-227. 

The methodology needs to be revised in a more organized way by adding a sub-heading for each step for clear understanding. 

It has been added the sub-heading 2.2.1 and 2.2.2 in the Materials and methods section

The picture/ illustration of the methodology should be expressed in a single composite image/ illustration following the sequence.

It has been expressed the figures 3 and 4 all in new figure 3.

Although the authors clearly mention the inclusion and exclusion criteria, the method of calculating the sample size is missing in the methodology

How the sample size calculation was performed is described in line 86-93.
"The sample size was calculated using Jamovi 2.3.18. As there were no previous studies comparing dental changes using BAME and MARPE, sample size calculations were based on the results of a pilot study performed in ten patients. The calculated mean ± standard deviation (SD) of the maxillary width change of BAME and MARPE was 1.96 mm ± 0.22 and 2.31 mm ± 0.35, respectively. Based on comparison of means, using two-tailed test, it was calculated that accepting an alpha risk of 0.05 and a beta risk of 0.2 in a bilateral contrast, 16 subjects per group are required to detect a difference equal to or greater than 0.35 mm. Finally, a sample was taken from 36 subjects"

The discussion section is in lacks citations and a logical explanation of the result obtained in this study.

The order of the discussions is described:

-Objectives of this study and the validation of the way to measure it.

-The significance of this study

- 2 examples of other studies (Jia et al and De Oliveira et al) and its relation with this study

-Explanation of our results

-Limitations of our study

-Clinical implications of our study.

The conclusion section needs to be revised with a more clear and summarized outcome of the study.

It has been changed the conclusions points to be more clear:

  • Patients treatedwith MARPE show torque increase in the teeth selected to support the expansion appliance, compared to cases treated with BAME.

    •  In cases where the upper molar torque was correct at the beginning of the treatment, a slight bone insertion gain was observed.

    •  In cases where the upper molar torque was negative or positive, a slight bone insertion loss was observed.

    •  Regarding the labial cortical bone loss, a slight gain of CBW was observed in all cases. This increase in labial cortical is higher in cases where posterior molar torque was negative before treatment.

It is attached the new version of the document and also the extra material of the CBCT measurements.

Round 2

Reviewer 2 Report

the authors have successfully satisfied all my previous concerns

sufficient